# Impact of Sea Warming and 17-α-Ethinylestradiol Exposure on the Lipid Metabolism of *Ruditapes philippinarum* Clams

**DOI:** 10.3390/ijms24119485

**Published:** 2023-05-30

**Authors:** João A. Rodrigues, Daniela S. C. Bispo, Mónica G. Silva, Rita Araújo, Amadeu M. V. M. Soares, Rosa Freitas, Ana M. Gil

**Affiliations:** 1CICECO—Aveiro Institute of Materials, Department of Chemistry, University of Aveiro, Campus Universitário de Santiago, 3810-193 Aveiro, Portugal; joao.rodrigues@ua.pt (J.A.R.); d.bispo@ua.pt (D.S.C.B.); anarita.asilva@ua.pt (R.A.); 2Department of Biology & Centre for Environmental and Marine Studies (CESAM), University of Aveiro, Campus Universitário de Santiago, 3810-193 Aveiro, Portugal; monicagsilva@ua.pt (M.G.S.); asoares@ua.pt (A.M.V.M.S.); rosafreitas@ua.pt (R.F.)

**Keywords:** bivalves, pharmaceutical drugs, global warming, lipids, metabolomics, nuclear magnetic resonance spectroscopy

## Abstract

This paper reports on an NMR metabolomics study of lipophilic extracts of *Ruditapes philippinarum* clams exposed to the hormonal contaminant 17-α-ethinylestradiol (EE2), at 17 °C and 21 °C. The results reveal that exposure at 17 °C triggers a weak response at low EE2 concentrations, suggestive of a slight increase in membrane rigidity, followed by lipid metabolic stability at higher EE2 concentrations. On the other hand, at 21 °C, lipid metabolism begins to respond at 125 ng/L EE2, with antioxidant docosahexaenoic acid (DHA) helping to tackle high-oxidative-stress conditions, in tandem with enhanced storage of triglycerides. Exposure to 625 ng/L EE2 (highest concentration) enhances phosphatidylcholine (PtdCho) and polyunsaturated fatty acid (PUFA) levels, their direct intercorrelation suggesting PUFA incorporation in new membrane phospholipids. This should lead to increased membrane fluidity, probably aided by a decrease in cholesterol. PUFA levels, considered a measure of membrane fluidity, were strongly (and positively) correlated to intracellular glycine levels, thus identifying glycine as the main osmolyte entering the cells under high stress. Membrane fluidity also seems to elicit the loss of taurine. This work contributes to the understanding of the mechanisms of response of *R. philippinarum* clams to EE2 in tandem with warming while unveiling novel potential markers of stress mitigation, namely high levels of PtdCho, PUFAs (or PtdCho/glycerophosphocholine and PtdCho/acetylcholine ratios) and linoleic acid and low PUFA/glycine ratios.

## 1. Introduction

Pharmaceutical compounds (PhACs) are contaminants of emerging environmental health concern which, if not removed efficiently from urban wastewaters by conventional wastewater treatment plants (WWTPs), may lead to adverse health effects on non-target aquatic wildlife [1,2,3]. Among PhACs, hormones are of particular interest due to their ability to alter endocrine system function [4]. For instance, the synthetic hormone 17-α-ethinylestradiol (EE2), used in formulations of oral contraceptives, hormone replacement therapies and breast and prostate cancer therapeutic protocols [5,6], has been increasingly studied due to its high estrogenic capacity, chemical stability and tendency to accumulate in aquatic biota, potentially reaching concentrations of up to ca. 100 ng/L [1,7]. Hence, EE2 has been included in the EU Watch List for water contaminants of emerging concern [8], and, indeed, exposure to EE2 has been reported to induce cellular and physiological negative effects in bivalves, such as oxidative stress in *Ruditapes philippinarum* clams [9]; oxidative stress and homeostasis imbalance in unionid *Lampsilis fasciola* mussels [10,11]; and increased metabolic capacity, cellular damage, loss of redox balance and histopathological damage in *Mytilus galloprovincialis* mussels [12,13]. In addition, the combined effects of EE2 toxicity and a warming scenario have been recently studied in marine invertebrates [9,12,14].

Oxidative stress is one of the early responses to chemical toxicity in marine organisms, which may lead to the oxidative damage of DNA, proteins and lipids [15]. Lipid peroxidation (LPO) levels are often used as an exposure toxicity biomarker, having been reported to change with steroid estrogen exposure [9,16] and temperature rise [12,17,18], possibly indicating activation of detoxification mechanisms, to minimize cellular damage and oxidative stress. Lipidomics (mostly mass spectrometry (MS)-based) has been extensively used for the lipid profiling of marine invertebrates, such as Cnidaria, Echinodermata, Arthropoda and Mollusca [19,20]. However, only a few studies have addressed chemical toxicity in marine invertebrates, such as bivalves, with examples found for exposure to metals [21,22], herbicides [23], crude oil and polycyclic aromatic hydrocarbons [24]. In alternative to MS methods, nuclear magnetic resonance (NMR) lipidomics has been more scarcely employed, mainly due to its lower sensitivity; however, NMR provides a more holistic view of different lipid families in a single experiment, higher reproducibility, along with more practical advantages such as sample minimal preparation and preservation. To our knowledge, only one NMR study of the lipid metabolism of bivalves has been reported [25] assessing the exposure of Sydney rock oysters, *Saccostrea glomerata*, to estrogenic mixtures. A significant reduction in phospholipids (PLs) and phosphatidylcholine (PtdCho) was associated with a reduction in available and/or an induction of reactive oxygen species (ROS) via estrogen metabolism, which may cause lipid peroxidation and lower availability of phospholipids. NMR lipidomics strategies have also been applied to study the effects of metal(loid)s on mosquitofish [26] and of diet interventions on the lipid content of fish [27,28].

In this study, untargeted ^1^H NMR was applied, for the first time to our knowledge, to measure the impact of temperature rise (from 17 °C as control temperature to 21 °C [29]), in tandem with exposure to EE2, on the lipid profile of *R. philippinarum* clams. This work builds on previous knowledge [14] that such conditions impact the osmoregulatory and energy metabolisms of these organisms enabling the articulation between polar [14] and lipophilic metabolomes, thus achieving a fuller metabolic description of the articulated effects of temperature rise and EE2 exposure on *R. philippinarum* clams.

## 2. Results

### 2.1. NMR Metabolic Spectra of R. philippinarum Clams’ Lipidic Extracts

Figure 1 shows a representative average ^1^H NMR spectrum of the lipidic extracts of clams, at 21 °C with no exposure to EE2 (control), visually similar to those recorded in other conditions of temperature (17 °C) and EE2 exposure. Peak assignment (Table 1) unveiled specific resonances from several lipid species, namely cholesterol (free and total), fatty acids (FAs) (including generally unsaturated FAs (UFAs), monounsaturated FAs (MUFAs), polyunsaturated FAs (PUFAs), e.g., ω-3 and ω-6 FAs, and in particular docosahexaenoic acid (DHA) and linoleic acid (LA)), PLs (PtdCho, PtdEtn, glycerophospholipids (GPLs) and plasmalogens), glycerolipids (mono- and triglycerides (MGs and TGs, respectively)) and sphingolipids (with the tentative assignment of two ceramide species at δ 5.54 (m) and 5.73 (br) (designated as ceramide 1 and 2, respectively). The predominating lipid families identified with NMR are broadly consistent with existing MS-based reports [19,20], which identified PLs, GPLs and FAs as the main lipid families in *Ruditapes philippinarum* species [30] and clams in general [19].

### 2.2. Adaptations of R. philippinarum Lipid Metabolome to EE2 Exposure at 17 °C and 21 °C

PCA and PLS-DA models based on the ^1^H NMR spectra of all samples exposed to EE2 at 17 °C or 21 °C showed large sample dispersion, with no clear group clustering or trajectory progression as a function of different EE2 concentrations. Indeed, sequential pairwise PLS-DA analysis (with gradually increasing EE2 concentrations, Appendix A) did not produce statistically robust models (note all Q^2^ < 0.3, Appendix A), thus indicating general low sensitivity of the lipidome of *R. philippinarum* clams to increasing EE2 concentrations at 17 °C. A similar behavior was observed at 21 °C, with the important exception of the 125 to 625 ng/L transition for which a possible group separation was observed, as Q^2^ = 0.54 (Appendix A). In addition, further pairwise PLS-DA models comparing each EE2 concentration with controls (Figure 2, Appendix A) suggested possible changes (Q^2^ = 0.3–0.4) occurring at 17 °C at lower EE2 concentrations (5 and 25 ng/L) and then at 625 ng/L (Q^2^ = 0.52). At 21 °C, changes compared to controls only seemed to occur at the highest concentration (Figure 2, left panel, Q^2^ = 0.51 and Appendix A). However, these hypotheses need to be confirmed/discarded through spectral integration (as discussed below).

Peak integration indicated that, at 17 °C, low EE2 concentrations (5 and 25 ng/L) induced slight decreases in FA methyls, GPLs and a ceramide species (Table 2 and Figure 3a, note the large errors), with a small increase in unassigned U1 (most probably arising from a cholesterol form) and an unidentified compound (comprising both unassigned peaks U3 and U4). At 125 and 625 ng/L, changes only affected the latter unknown compound. At 21 °C, the lipid metabolome was confirmed not to respond to low EE2 concentrations, with 125 ng/L EE2 inducing a significant decrease in DHA and an increasing tendency for TGs (Table 2, Figure 3b). The highest EE2 concentration, 625 ng/L, led to a stronger response in bivalves’ lipidome (Table 2, Figure 3b), comprising significant increases in several unsaturated FAs (specifically ω-3 and ω-6 FAs, including LA), also viewed through general UFA and PUFA resonances. Concomitantly, increases occurred in PtdCho, GPLs in general and a ceramide species, while unspecified cholesterol species (corresponding to resonances U1 and U2) were decreased (Table 2). Notably, the unsaturated FA increase at 625 ng/L EE2 was not marked enough to increase the average polyunsaturation degree, as measured through the NMR spectra.

Considering that saturated FAs (viewed through the (CH_2_)_n_ resonance at δ 1.27 ppm, Appendix A) are expected to serve as the main substrates for storage fat in the form of TGs, the TG/FA (CH_2_)_n_ ratio was used to investigate such conversion. This ratio tended to increase at 125 ng/L EE2, particularly at 21 °C (Figure 4a, left and middle panels), reflecting the marked TG increase (Figure 3b). Good TG/sat. FA correlations were found at most EE2 concentrations at 21 °C but not at 17 °C (Figure 4a, right panel and Appendix A), thus confirming that TG storage occurs at the expense of saturated FAs, although to higher extension at 125 ng/L EE2 (TG increase, Figure 3b). At 625 ng/L EE2, the decreases in TGs (Figure 3b) and TGs/sat.FAs (Figure 4a) suggest a slowing down of lipid storage. Interestingly, no TG vs. saturated FA correlations were seen at 17 °C (Appendix A). The increase in unsaturated FAs at 21 °C and 625 ng/L EE2 (Figure 3b) occurred concomitantly with increases in PtdCho and a ceramide (also a membrane lipid, with particular importance in signaling processes), in tandem with a decrease in unspecified cholesterol forms (also probably linked to membrane composition). This indicates the importance of membrane remodeling metabolism at 21 °C for high EE2 concentrations, as reported with a basis on polar extract metabolomics [14]. The PtdCho/PtdEtn ratio remained practically unchanged at both temperatures (Figure 4b), meaning that the PtdCho increase at 21 °C and 625 ng/L EE2 (Figure 3b) is accompanied by an increase in PtdEtn, although not statistically significant (as it originated a weak and broad NMR resonance, Figure 1). Together with the good PtdCho vs. PtdEtn correlations (r = 0.58–0.73) at all EE2 concentrations and at both temperatures (Figure 4b, right panel for 625 ng/L EE2, and Appendix A), this result indicates that membrane composition in terms of PtdCho and PtdEtn relative levels seems independent of EE2 concentration or temperature.

We hypothesize that enhanced PtdCho synthesis (particularly at 21 °C and 625 ng/L EE2, Figure 3b) may be preferentially involving unsaturated FAs, and the unchanged PtdCho/PUFAs ratios at both temperatures (Figure 4c) suggest that enhanced PtdCho levels (Figure 3b) are accompanied by enhanced PUFA synthesis. Indeed, the strong correlation between the two lipid species, at all EE2 concentrations and both temperatures (r = 0.85–0.93, Appendix A), confirms this underlying relationship. The significant enhancements of both PtdCho and PUFA levels at 21 °C and at 625 ng/L EE2 (Figure 3b) suggest that extreme stressor conditions enhance both biosynthetic processes. Other PtdCho precursors, namely acetylcholine and GPC, were seen to decrease in polar extracts at 21 °C and 625 ng/L EE2 (Figure 5, right panel) [14]. PtdCho/precursor ratios (Figure 5) (making use of the levels measured in polar extracts, so that only ratio variations should be considered rather than ratio absolute values) show relevant increases at 21 °C for PtdCho/GPC and PtdCho/acetylcholine (but not for PtdCho/choline), in tandem with decreases in GPC and acetylcholine in the corresponding polar extracts (Figure 5, right panel). This indicates that PtdCho synthesis relies importantly on acetylcholine and GPC pools, particularly at 21 °C and 625 ng/L of EE2. Hence, the 21 °C/625 ng/L EE2 treatment seems to lead to the enhancement of PtdCho biosynthesis at the expense not only of PUFAs but also of GPC and acetylcholine, in particular.

To investigate the potential relationship between membrane composition, fluidity and osmolyte transport fluxes, the ratios of PUFAs/osmolytes were calculated, again using the levels measured in polar extracts [14] for selected osmolytes (namely glycine, homarine, taurine and trigonelline, the increased levels of which, at 21 °C and 625 ng/L EE2, may result from osmoregulation adaptations and/or protective antioxidant mechanisms [32,33]. Indeed, assuming that one of the main fates of PUFAs is their incorporation in membrane PLs (namely PtdCho), membrane fluidity should become higher (particularly at 21 °C and 625 ng/L EE2), and this should affect osmolyte transport across cell membranes. The ratios of PUFAs/osmolytes calculated at 21 °C (Figure 6, left panel) were generally maintained constant throughout the EE2 concentration range, except for a relevant decrease in PUFAs/glycine and an increase in PUFAs/taurine at 625 ng/L (Figure 6a,b, left panel), reflecting the reported high glycine and taurine levels (Figure 6a, middle) [14]. The strongest positive PUFAs vs. glycine correlation at 21 °C and 625 ng/L (r = 0.76) (Figure 6a, right panel) (while present at all EE2 concentrations at 21 °C but not at 17 °C, Appendix A) confirmed that, at 21 °C, glycine transport into the cells is aided by higher PUFA-enhanced membrane fluidity, becoming particularly extensive at 625 ng/L EE2. In the case of taurine, PUFAs vs. taurine correlation is only seen at 21 °C and 625 ng/L (Figure 6b, left panel, and Appendix A), following a negative trend. Conversely, homarine (Figure 6c) and trigonelline, also reported to increase at the highest stressor conditions [14], showed no meaningful correlations with PUFAs (r < 0.40).

## 3. Discussion

### 3.1. Exposure to EE2 at 17 °C

The multivariate results obtained for *Ruditapes philippinarum* lipid extracts generally reflected large sample dispersion and weak group distinction, which contrasted with the behavior reported for the polar extracts of the same samples [14], thus indicating fewer changes occurring in the bivalves’ lipidome (as viewed with NMR). However, it was possible to detect distinct dynamics of response to EE2 between 17 °C and 21 °C, similarly to observations based on the polar metabolome. Indeed, at 17 °C, exposure to 5 and 25 ng/L EE2 induced a weak response in compounds related to cell membranes (decreased GPL and ceramide and increased cholesterol species), an effect that became attenuated at higher EE2 concentrations. This accompanies the reported hormesis behavior at 17 °C viewed through the polar metabolome [14], which showed that low EE2 concentrations lead to energy metabolism activation and a decrease in acetylcholinesterase (AChE) activity [9,14]. The AChE activity decrease at lower EE2 concentrations is likely to deplete choline pools and, in turn, decrease GPL as noted here. GPL decrease has also been suggested to reflect lower availability of stored lipids (TGs) [21]; however, this does not apply at low EE2 concentrations, as TG levels increase (qualitatively) up to 125 ng/L EE2. PLs have also decreased in bivalves exposed to estrogen mixtures at 22 °C, having been related to reactive oxygen species (ROS)-caused lipid oxidation [25]. However, this is not consistent with GPL decreasing at lower oxidative stress levels (low EE2 concentrations) and then remaining unchanged at the higher stress levels caused by higher EE2 concentrations [9]. We hypothesize therefore that GPL decreases at 17 °C and low EE2 concentrations due to lower AChE activity and probably to promote some extension of membrane remodeling. The concomitant increase in cholesterol species suggests that membrane rigidity may be slightly enhanced under such conditions (since cholesterol content is expected to correlate directly with membrane rigidity [34]), as a protective mechanism effective at lower temperatures. The tendency for ceramide reduction has been related to stress regulation processes [35,36], although at the lower stress levels characterizing 17 °C and low EE2 concentrations, the origin for such tendency requires further investigation.

### 3.2. Exposure to EE2 at 21 °C

At 21 °C, the lipid metabolome of *R. philippinarum* does not exhibit any low EE2 concentration effects and only begins to respond significantly at 125 ng/L EE2 (with decreased DHA and increased TGs). DHA decrease is consistent with higher oxidative stress levels measured at higher EE2 concentrations [9], as this highly unsaturated essential FA (22:6, ω-3) is a recognized potent antioxidant, capable of accelerating the action of enzymatic and nonenzymatic antioxidants and reducing ROS accumulation, as shown for *R. philippinarum* clams exposed to long-term daily rhythms of air [37]; furthermore, DHA is also able to down-regulate the expression of pro-inflammatory mediators related to cytotoxic cell damage and up-regulate the expression of anti-inflammatory agents [38]. We hypothesize that DHA is mobilized at 125 ng/L EE2, at 21 °C, to enhance the bivalve antioxidant response, together with other antioxidant metabolites such as taurine/hypotaurine, homarine and trigonelline [14] (Figure 7). Interestingly, the role of DHA seems to be attenuated or rendered insufficient to face the higher oxidative stress induced by 625 ng/L EE2. A second feature characterizing the effects of 125 ng/L EE2 exposure at 21 °C regards storage lipids, as viewed by TGs. Interestingly TG levels creep up with EE2 concentration and peak at 125 ng/L, only to decrease again at 625 ng/L. We have established that TG storage at the expense of saturated FAs is an ongoing process at any EE2 concentration at 21 °C (and not at 17 °C, probably where protein and glycogen reserves suffice cellular needs [9]). However, TG storage is intensified at 125 ng/L (Figure 7) perhaps to better preserve energy sources, subsequently decreasing at 625 ng/L EE2. Such decrease is consistent with the previous suggestion that lipids are used to produce minimal energy levels, under such conditions [14], while glycogen and protein reserves are enhanced [9], the latter probably reflecting the more extensive biosynthesis of protective enzymes (such as GPx and GSTs) at 125 and 625 ng/L [9].

Undoubtedly, it is the 625 ng/L EE2 concentration that triggers the strongest response in bivalves’ lipidome at 21 °C (Figure 7). We propose that important membrane modulation/degradation processes occur, with enhanced PtdCho, ceramide and PUFA levels, PtdCho synthesis using PUFAs (ω-3 and ω-6 FAs and LA, in particular), acetylcholine and GPC as precursors. Bivalves may be synthesizing higher PUFA levels to exploit their antioxidative properties to fight high stress levels; however, PUFA integration in membrane PLs also seems to occur. This will lower membrane transition temperature and increase membrane fluidity, thus leading to less stable membranes. Indeed, high amounts of PUFAs have been shown to promote high cell membrane permeability in bivalves [39,40], thus determining the organisms’ capacity to tackle environmental and chemical stressors [40]. Importantly, the concomitant decrease in cholesterol content may also contribute to higher membrane fluidity. Elevated phospholipid content (here expressed as a PtdCho increase) has indeed been proposed to help keep membranes permeable and capable of cellular volume regulation, in response to low seawater salinity as a stressor [41], although we propose that temperature and PL composition will determine the exact endpoint of cell stability. Other aspects of membrane organization, e.g., PtdCho/PtdEtn ratio, or plasmalogen proportion, can regulate membrane function in addition to changes in lipid unsaturation and overall fluidity [40]. In this work, the PtdCho/PtdEtn ratio remained constant throughout the range of EE2 concentrations under study, and no significant changes were observed in plasmalogens.

The above-discussed membrane fluidity increase was here demonstrated to lead to a large increase in intracellular levels of glycine (the osmolyte that more strongly correlated (positively) with membrane fluidity, as viewed by PUFA levels), compared to taurine, homarine or trigonelline. Glycine is one of the key FAAs involved in osmoregulation and cell volume adjustment in hard clams, with reports describing its continuous increase under hypersalinity stress [30], and we propose that this amino acid is the main osmolyte entering the cells in *R. philippinarum* clams exposed to EE2 at 21 °C. The absence of meaningful correlations between membrane fluidity and the increasing levels of homarine or trigonelline suggest that these compounds are either naturally present in cells (not needing to cross the membrane) and/or are involved in a variety of roles (which would diffuse any existing correlation with membrane characteristics). Homarine has been reported as important to regulate ionic strength and maintain intracellular osmotic pressure [42,43], while trigonelline has been identified as a major metabolite in the osmotic regulation of marine mollusks [43,44], being known to increase antioxidant enzyme activity and decrease lipid peroxidation in mammals [45]. The role of taurine seems to be more complex, as it exhibited a negative (*r* ca. 0.5) correlation with PUFA levels (membrane fluidity). This would suggest that taurine might be crossing the membrane to exit cells, and, indeed, taurine levels decrease at 625 ng/L compared to 125 ng/L EE2 which suggests either its loss due to membrane degradation and/or its lesser production through hypotaurine oxidation, although the latter explanation is somewhat inconsistent with the high oxidative stress levels reported for 625 ng/L EE2 at 21 °C [9,14]. Taurine has been seen to increase with the exposure of *Haliotis diversicolor* shellfish (muscle/liver) to hypoxia and warming [33], similarly to the 125 ng/L EE2/21 °C conditions in the present study. At 625 ng/L, we hypothesize that taurine is most probably lost from the cytosol due to extensive membrane fluidity or even degradation.

## 4. Materials and Methods

### 4.1. Sampling and Experimental Conditions

Sampling protocols and experimental conditions have been described in detail elsewhere [9,14]. Clams *Ruditapes philippinarum* were collected in the Ria de Aveiro (northwest Atlantic coast of Portugal) in September 2019, and organisms with similar size (length: 3.81 ± 0.42 cm; width: 3.06 ± 0.51 cm) were selected for the study. For depuration and acclimation to laboratory conditions, clams were kept in artificial seawater for 10 days (salinity 30 ± 1, Tropic Marin^®^ SEA SALT from Tropic Marine Center), at 17 ± 1 °C, under continuous aeration and a natural photoperiod. The artificial seawater was renewed every 2−3 days. After this period, the clams were subjected to a 28-day chronic toxicity test (Supporting Information, Appendix A). Five different EE2 concentration levels were tested (Sigma-Aldrich, purity ≥ 98%, M_W_ = 296.40 g/mol, 1 mg/L stock solution in ultrapure water): 0 (control group), 5, 25, 125 and 625 ng/L. To evaluate the combined effects of temperature rise and EE2 exposure, two different water temperatures were considered: 17 ± 1 °C (control, mean temperature of Ria Aveiro during September [46]) and 21 ± 1 °C (a 4 °C temperature rise, corresponding to the worst-case climate change scenario [29]). For each treatment (temperature/EE2 concentration), 12 individuals were considered (3 aquaria per treatment, 4 individuals per aquarium). The 3 L glass aquaria were placed in distinct climatic rooms for each temperature (for the assays at 21 °C, the temperature was raised by 2 °C every 2–3 days until the final temperature was reached). During the experimental period, all aquaria were maintained under continuous aeration and a natural photoperiod. During EE2 exposure, the water from each aquarium was renewed once a week, after which EE2 concentrations were re-established. Individuals were fed with Algamac Protein Plus (150,000 cells/animal/day) every 2–3 days. Mortality was checked daily. At the end of the exposure period, each clam was homogenized, frozen in liquid nitrogen and stored at −80 °C.

### 4.2. Tissue Extraction Procedure

The detailed clam metabolite extraction procedure has been described before [14], following the method based on solvent extraction using methanol, water and chloroform [47]. The method involved the grinding of the clams’ whole soft tissue (0.15 g/sample fresh weight), with a pestle and mortar, in liquid nitrogen. The ground samples were transferred to microtubes, followed by addition of cold methanol (600 µL), ultrapure water (128 µL) and chloroform (300 µL). The resulting mixture was vortexed, left in ice for 10 min and then centrifuged (2500× *g*, 4 °C, 10 min). The supernatant was transferred into a microtube to which 300 µL of chloroform and 300 µL of water were added. The mixture was vortexed, centrifuged again (2500× *g*, 4 °C, 10 min) and left in ice for 5 min. The bottom (apolar) layer was collected and transferred into glass amber vials, dried under a nitrogen flow, and stored at −80 °C until NMR analysis.

### 4.3. NMR Spectroscopy

The dried lipidic extracts were re-suspended in 600 µL of deuterated chloroform (99.8% deuterium, Eurisotop D307F) containing 0.03% tetramethylsilane (TMS) for chemical shift referencing. After vortex homogenization, 550 μL of solution was transferred to 5 mm NMR tubes. For each sample, one unidimensional (1D) 1H NMR spectrum was acquired on a Bruker AVANCE III 500 spectrometer (Bruker, Rheinstetten, Germany), operating at 500.13 MHz for 1H observation, using a 5 mm inverse probe, at 298 K, using a standard 90° pulse sequence, “zg” pulse sequence (Bruker library). A total of 512 scans were collected into 32 k data points, with a spectral width of 7002.8 Hz, acquisition time of 2.34 s and relaxation delay of 3 s. Prior to Fourier transformation, each free-induction decay was zero-filled to 64 k points and multiplied by a 0.3 Hz exponential line-broadening function. The spectra were manually phased, baseline corrected, and chemical shift referenced to TMS signal, at δ = 0.00 ppm. Peak assignments were based on existing literature [48,49,50,51], spiking experiments (specifically for PtdCho, phosphatidylethanolamine (PtdEtn), sphingomyelins (SMs), mono- and triglicerides (MGs, TGs), free and esterified cholesterol (Chol., EChol.), arachidonic acid (AHA), oleic acid (OA) and docosahexaenoic acid (DHA)) and spectral databases (namely the Biological Magnetic Resonance Data Bank (BMRB) [52] and the human metabolome database (HMDB)) [53]. Statistical Total Correlation Spectroscopy (STOCSY) [54] was also used to aid assignment.

### 4.4. Statistical Analysis of NMR Spectra

Prior to multivariate analysis, full-resolution 1D ^1^H NMR spectra were converted to data matrices (AMIX-viewer 3.9.14, BrukerBiospin, Rheinstetten, Germany), aligned using the recursive segment-wise peak alignment method [55] (Matlab 8.3.0, The MathWorks Inc., Natick, MA, USA) and normalized to spectral total area, excluding the water (δ 1.50–1.57 ppm) and chloroform (δ 7.00–7.50 ppm) regions. Unsupervised principal component analysis (PCA) [56] and supervised partial-least-squares discriminant analysis (PLS-DA) [57] models were performed on the ^1^H NMR data, upon unit variance (UV) scaling (SIMCA-P 11.5; Umetrics, Umeå, Sweden). The obtained models were considered satisfactorily robust for predictive power (Q^2^) values approaching or above 0.5. For univariate analysis, each identified metabolite or relevant resonance peaks were integrated into the raw spectra (Amix 3.9.5, Bruker BioSpin, Rheinstetten, Germany) and normalized to the total spectral area (excluding the water and chlorofom regions). For each pairwise comparison, effect size (ES) values [31,58] and statistical significance (*p*-values) were assessed by the Wilcoxon test (R software, version 4.0.0, combined with R-studio version 1.3.1093, R Foundation for Statistical Computing, Vienna, Austria). Benjamini–Hochberg false discovery rate (FDR) [59] correction was applied for multiple testing. The average fatty acid chain length and degree of polyunsaturation were calculated as described in [60]. Boxplots were built using “ggplot2” package, R software [61]. STOCSY analysis [54] was performed on the normalized data (Matlab 8.3.0, The MathWorks Inc., Natick, MA, USA) for peak assignment.

## 5. Conclusions

We report on the adaptations of the lipid metabolism of *Ruditapes philippinarum* clams, when exposed to 17-α-ethinylestradiol (EE2), under warming conditions (17 °C and 21 °C), using NMR metabolomics. Whereas the control temperature only triggers a weak response at low EE2 concentrations, subsequently reaching lipid metabolic stability (at higher EE2 concentrations), when the temperature is raised to 21 °C, lipid metabolism begins to respond significantly at 125 ng/L EE2, with selected PUFAs playing an antioxidant role, while enhanced storage of triglycerides at the expense of saturated FAs is noted. The highest EE2 concentration tested (625 ng/L) induces increases in PtdCho and PUFA biosynthesis, probably leading to increased membrane fluidity, aided by a decrease in cholesterol. This effect was found to strongly correlate with intracellular glycine levels, identifying this amino acid as the main osmolyte entering the cells under high stress. Other osmolytes seem to behave differently, with the apparent loss of taurine due to membrane instability, while homarine and trigonelline exhibited no clear dependence on membrane fluidity.

As well as contributing to the understanding of the response mechanisms of *R. philippinarum* clams exposed to EE2 and warming conditions, this work unveils novel potential markers of stress mitigation mechanisms, such as high levels of PtdCho and PUFAs (or more specifically, PtdCho/GPC, PtdCho/acetylcholine ratios and LA), as well as low PUFA/glycine ratios. Follow-up studies will investigate the relationship between these biochemical changes and potential mechanisms of *R. philippinarum* transformation and survival under environmental stress.

## Figures and Tables

**Figure 1 ijms-24-09485-f001:**
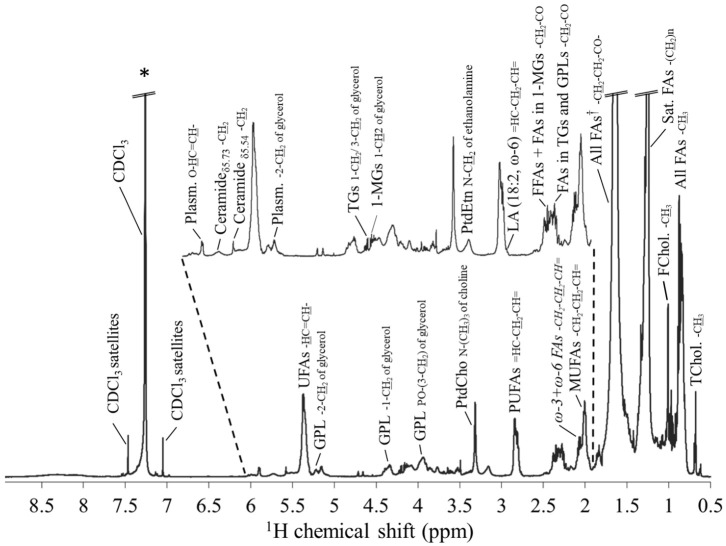
A 500 MHz average ^1^H NMR spectrum of bivalve lipidic extracts corresponding to control conditions (no EE2 exposure), at 21 °C. * Resonance arising from chloroform (δ 7.5–7.0) and excluded from multivariate analysis matrix; †: overlapping of residual water resonance and that of -CH_2_-CH_2_-CO- protons from fatty acids (δ 1.8–1.4). Abbreviations: 1-MGs, 1-monoacylglicerides; FAs, fatty acids; FChol., free cholesterol; FFAs, free FAs; GPLs, glycerophospholipids; MUFAs, monounsaturated FAs; Plasm., plasmalogens; PtdCho, phosphatidylcholine; PtdEtn, phosphatidylethanolamine; PUFAs, polyunsaturated FAs; Sat. FAs, saturated FAs; TChol., total cholesterol; TGs, triacylglycerides; and UFAs, unsaturated FAs. A full list of assignments may be found in Table 1.

**Figure 2 ijms-24-09485-f002:**
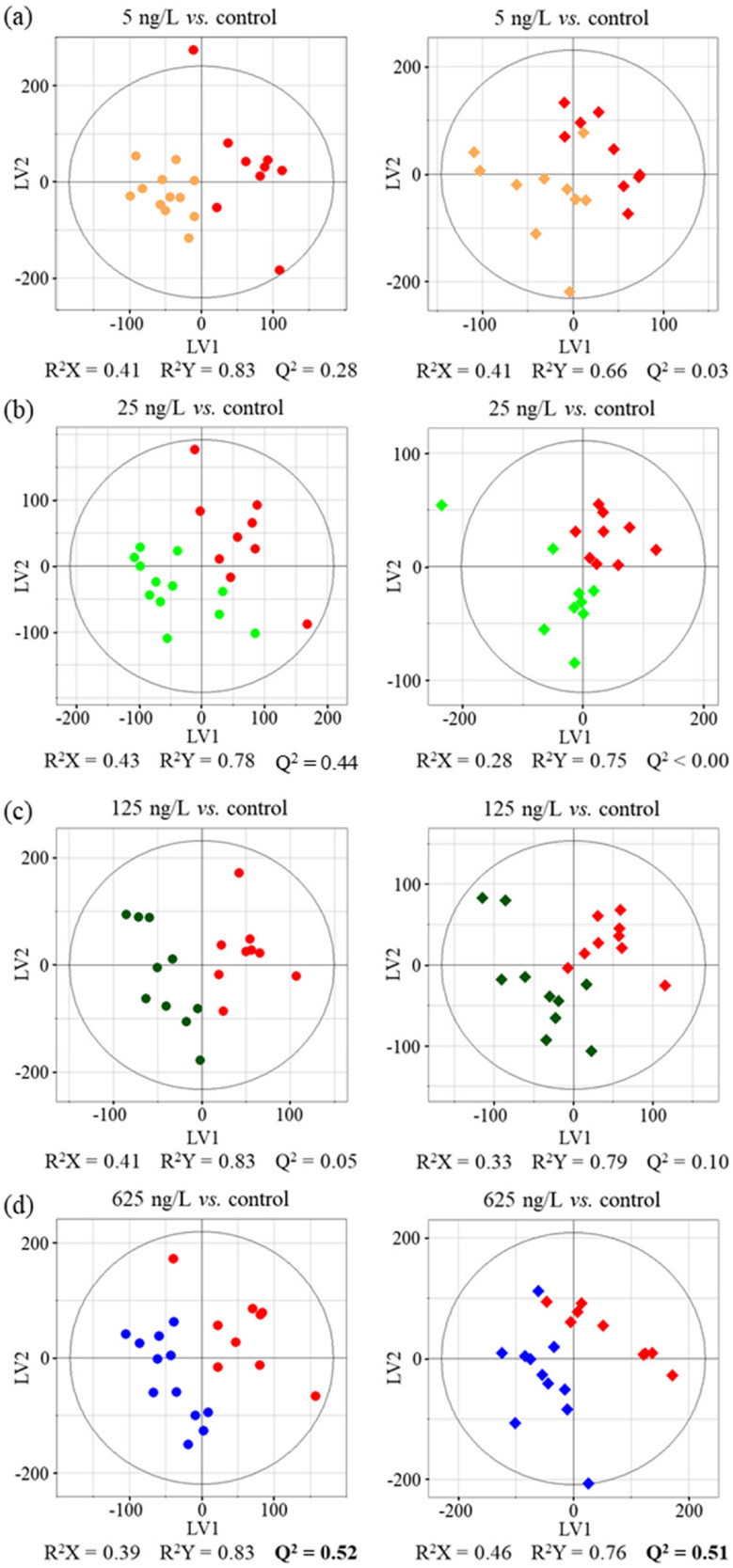
Pairwise PLS-DA score plots obtained for the ^1^H NMR spectra of bivalves’ lipid extracts, upon EE2 exposure at 17 °C (**left**) and at 21 °C (**right**), compared to controls. (**a**) 5 ng/L (orange) vs. 0 ng/L (red); (**b**) 25 ng/L (light green) vs. 0 ng/L (red); (**c**) 125 ng/L (dark green) vs. 0 ng/L (red) and (**d**) 625 ng/L (blue) vs. 0 ng/L (red). LV, latent variables; R^2^X: fraction of the variation of X-variables explained by the model; R^2^Y: fraction of the variation of Y-variables explained by the model; Q^2^, predictive power (Q^2^ > 0.50 in bold).

**Figure 3 ijms-24-09485-f003:**
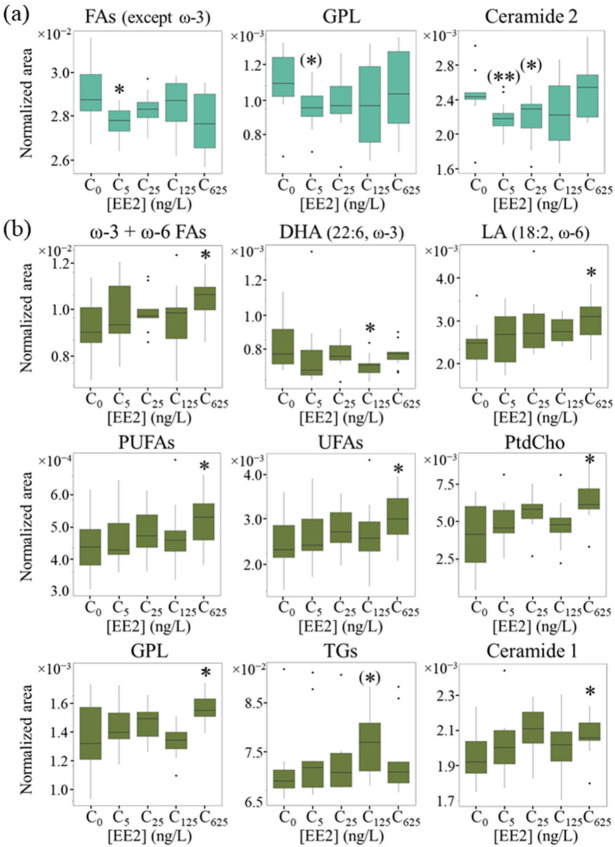
Boxplots of normalized integrals of lipids identified as relevant varying (*p* < 0.05) as a function of EE2 concentration, (**a**) at 17 °C (light green), namely FAs (except ω-3, -CH_3_), GPLs, PO-(3-CH_2_) and ceramide 2 at δ 5.73, and (**b**) at 21 °C (dark green), namely ω-3 + ω-6 FAs; DHA, LAs, PUFAs, UFAs, PtdCho; GPLs, TGs and ceramide 1 at δ 5.54 (all abbreviations as defined in Figure 1 and Table 1). * *p* < 0.05 and ** *p* < 0.01 express statistical relevance compared to controls. (*) signals with effect size (ES) error > ES (but *p* < 0.05).

**Figure 4 ijms-24-09485-f004:**
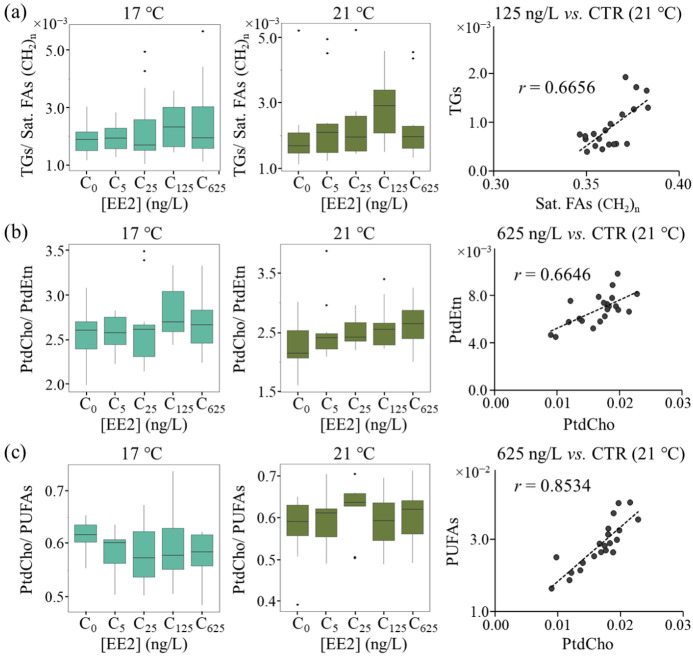
Variations in selected lipid ratios as a function of EE2 concentration and at both 17 °C (**left,** light green) and 21 °C (**middle**, dark green) and corresponding correlation plots (r) for selected EE2 concentration/temperature conditions (**right**). (**a**) TGs/saturated FAs, (CH_2_)_n_; (**b**) PtdCho/PtdEtn; and (**c**) PtdCho/PUFAs. Abbreviations: CTR: control group; other abbreviations as defined in Figure 1 and Table 1.

**Figure 5 ijms-24-09485-f005:**
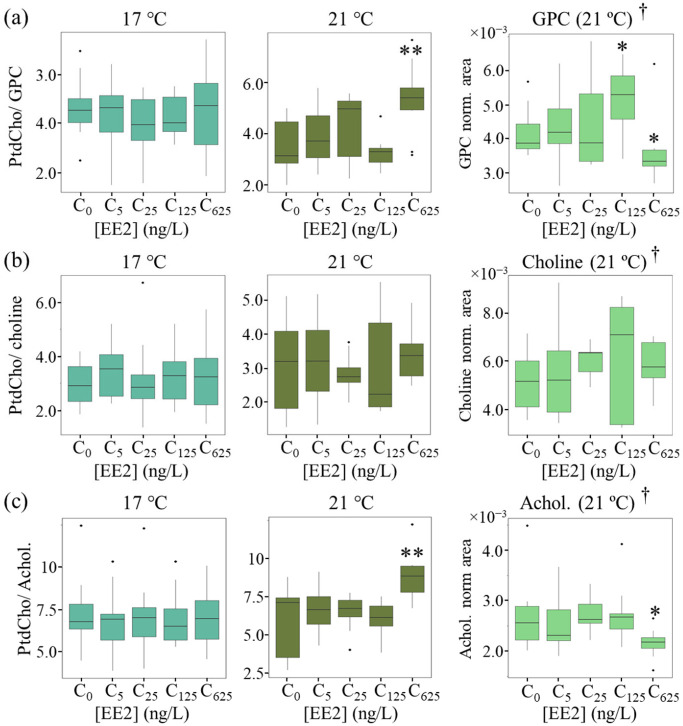
Ratios of PtdCho and corresponding potential polar precursors, as a function of EE2 concentration and temperature. PtdCho ratios with (**a**) GPC, (**b**) choline and (**c**) acetylcholine, at 17 °C (**left**, light green) and 21 °C (**middle**, dark green); and precursor variations in polar extracts, at 21 °C (**right**, bright green) [14]. † Measured in polar extracts [14]. Abbreviations: Achol., acetylcholine; GPC, glycerophosphocholine; other abbreviations as defined in Figure 1 and Table 1. * *p* < 0.05 and ** *p* < 0.01 express statistical relevance compared to controls.

**Figure 6 ijms-24-09485-f006:**
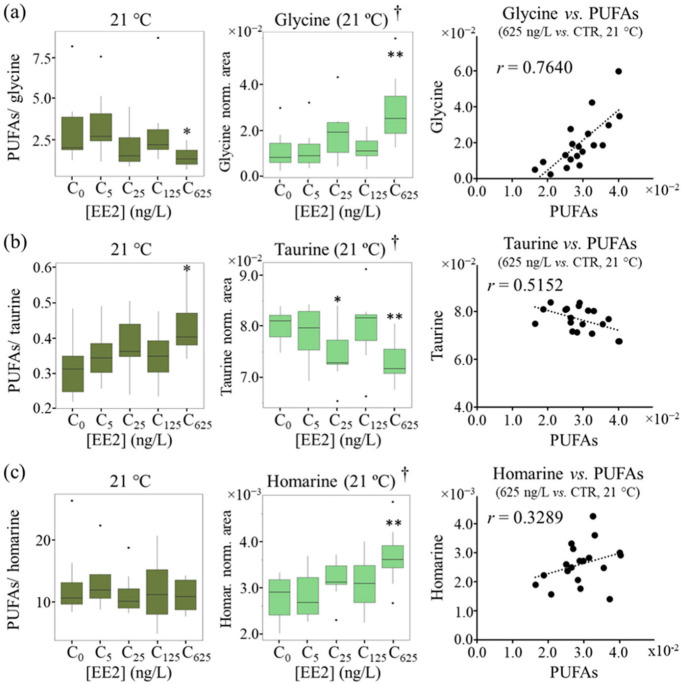
Variations in selected ratios between PUFAs (lipid extracts) and osmolytes (polar extracts) glycine, taurine and homarine, at 21 °C, as a function of EE2 concentration. Boxplots of PUFAs/osmolyte ratios (**left**, dark green), osmolyte levels in polar extracts (**middle**, bright green) [14] and correlation plot (r) (**right**) between PUFAs and osmolytes, for osmolytes (**a**) glycine, (**b**) taurine and (**c**) homarine. Abbreviations as defined in Figure 1 and Table 1. † Metabolite variations measured in polar extracts [14]. * *p* < 0.05 and ** *p* < 0.01 express statistical relevance compared to controls.

**Figure 7 ijms-24-09485-f007:**
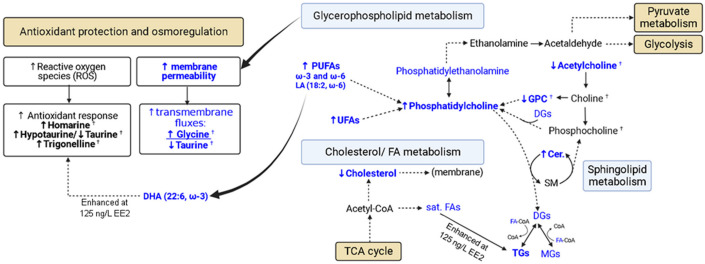
Putative relevant metabolic alterations taking place at 625 ng/L EE2 and 21 °C (in royal blue) unless otherwise stated, mainly with basis on lipidic compound variations. Metabolites identified as altered are shown in bold. Thin arrows: solid and dashed arrows correspond to single and multiple steps in the pathway, respectively. Dark wider arrows express qualitative relationships Vertical up/down arrows preceding a metabolite name indicate the direction of variation in the levels of that metabolite. † metabolites measured in polar extracts [14]. Abbreviations: CoA, coenzyme A; DGs, diacylglycerides; SM, sphingomyelins; other abbreviations as defined in Figure 1 and Table 1.

**Table 1 ijms-24-09485-t001:** Assignment of lipid resonances identified in the ^1^H NMR (500 MHz) spectra of bivalves’ lipophilic extracts organized into families and listed by increasing chemical shift (in each family). Underlined protons (right column) correspond to those for which chemical shift is indicated. Abbreviations: DHA, docosahexaenoic acid; FAs, fatty acids; FFAs, free FAs; GPLs, glycerophospholipids; LA, linoleic acid; MG, monoacylgliceride; MUFAs, monounsaturated FAs; PtdCho, phosphatidylcholine; PtdEtn, phosphatidylethanolamine; PUFAs, polyunsaturated FAs; TGs, triacylglycerides; UFAs, unsaturated FAs; s, singlet; d, doublet; dd, doublet of doublets; ddd, doublet of doublets of doublets; t, triplet; m, multiplet; br, broad signal. ^a^ tentative assignment.

Compound		δ ^1^H, ppm (Assignment, Multiplicity)
Cholesterol	Total	0.68 (s, 18-CH_3_)
	Total	0.86 (d, 26-CH_3_)
	Total	0.87 (d, 27-CH_3_)
	Total	0.91 (d, 21-CH_3_)
	Free	1.01 (s, 19-CH_3_)
	Total	1.12 (m, multiple cholesterol protons)
	Total	1.49 (m, multiple cholesterol protons)
	Total	1.84 (m, multiple cholesterol protons)
	Total	1.96 (br, 7-CH_2_/8-CH)
	Total	2.23 (br, 4-CH_2_)
	Free	3.53 (br, 4-CH_2_)
Fatty acids (FAs)	All FAs (except ω-3)	0.88 (t, CH_3_)
	ω-3 FAs	0.97 (t, CH_3_)
	Saturated FAs	1.27 (m, (CH_2_)n)
	All FAs	1.60 (m, -CH_2_-CH_2_-CO-)
	All UFAs	1.98–2.09 (m, -CH_2_-CH_2_-CH=)
	MUFAs	2.01 (m, -CH_2_-CH_2_-CH=)
	ω-3 + ω-6 FAs	2.07 (m, -CH_2_-CH_2_-CH=)
	FAs in TGs and GPLs	2.29 (m, -CH_2_-CO)
	FFAs + FAs in MGs	2.35 (t, -CH_2_-CO)
	DHA (22:6, ω-3)	2.42 (m, -CH_2_-CH_2_-CO)
	LA (18:2, ω-6)	2.77 (t, =CH-CH_2_-CH=)
	PUFAS	2.83 (m, =CH-CH_2_-CH=)
	UFAs	5.36 (m, -HC=CH-)
Glycerophospholipids (GPLs)	PtdEtn	3.16 (br, N-CH_2_)
	PtdCho	3.32 (s, -N^+^(CH_3_)_3_)
	PtdCho	3.74 (br, N-CH_2_)
	Plasmalogens	3.85 (m, 1-CH_2_ glycerol)
	All GPL	3.95 (m, PO-(3-CH_2_) glycerol)
	PtdEtn	4.06 (br, PO-CH_2_ ethanolamine)
	All GPL	4.35 (m, 1-CH_2_ glycerol)
	Plasmalogens	5.16 (m, 2-CH_2_ glycerol)
	All GPL	5.22 (m, 2-CH glycerol)
	Plasmalogens	5.90 (d, O-CH=CH)
Glycerolipids	1-MGs	3.65 (ddd, 3-CH_2_ glycerol)
	TGs	4.15 (dd, 1-CH_2_/3-CH_2_ glycerol)
	1-MGs	4.18 (ddd, 1-CH_2_ glycerol)
	TGs	4.29 (dd, 1-CH_2_/3-CH_2_ glycerol)
	2-MGs	4.93 (m, 2-CH glycerol)
	TGs	3.65 (m, 2-CH glycerol)
Sphingolipids	Ceramide 1 ^a^	5.54 (m, -CH_2_)
	Ceramide 2 ^a^	5.73 (br, -CH_2_)

**Table 2 ijms-24-09485-t002:** Lipid compound variations (*p* < 0.05) identified in the lipid extracts of bivalves exposed to different EE2 concentrations (5, 25, 125 and 625 ng/L) compared to control samples (0 ng/L), at (i) 17 °C (top section) and (ii) 21 °C (bottom section); all FDR corrected values had *p*-value > 0.05. Underlined protons (right column) correspond to those for which chemical shift is indicated. ^a^ integrated peak within spin system; ^b^ effect size (ES) and corresponding error calculated as described previously [31]; ^c^ significance level 95% (*p* < 0.05); ^d^ tentative assignment. ^e^ probably arising from cholesterol species; ^f^ possibly the same spin system, as shown by high STOCSY correlation. Values in brackets correspond to values where ES error > ES. U1–4: unassigned resonance 1-4; glyc.: glycerol.

Metabolites	δ H, ppm (Multiplicity) ^a^	5 ng/L vs. 0 ng/L		25 ng/L vs. 0 ng/L		125 ng/L vs. 0 ng/L		625 ng/L vs. 0 ng/L	
		ES ± Error ^b^	*p*-Value ^c^	ES ± Error ^b^	*p*-Value ^c^	ES ± Error ^b^	*p*-Value ^c^	ES ± Error ^b^	*p*-Value ^c^
**EE2 exposure at 17 °C**									
All FAs (except ω-3)	0.88 (t, CH_3_)	−1.0 ± 0.9	3.3 × 10^−2^	---	---	---	---	---	---
All GPLs	3.95 (m, PO-(3-CH_2_))	(−0.8 ± 0.9)	(3.9 × 10^−2^)	---	---	---	---	---	---
Ceramide 2 δ 5.73 ^d^	5.73 (br, CH_2_)	(−0.8 ± 0.9)	(3.6 × 10^−3^)	(−0.7 ± 0.9)	(2.8 × 10^−2^)	---	---	---	---
U1 ^e^	0.84 (br, weak)	1.0 ± 0.9	2.3 × 10^−2^	---	---	---	---	---	---
U3 ^f^	2.88 (s, weak)	−1.2 ± 0.9	2.2 × 10^−3^	−1.6 ± 1.0	2.2 × 10^−4^	---	---	−1.2 ± 0.9	4.5 × 10^−3^
U4 ^f^	2.96 (s, weak)	−1.0 ± 0.9	3.6 × 10^−3^	−1.4 ± 0.9	3.8 × 10^−4^	−1.1 ± 0.9	3.8 × 10^−4^	−1.1 ± 0.9	2.8 × 10^−3^
**EE2 exposure at 21 °C**									
ω-3 + ω-6 FAs	2.07 (m, -CH_2_-CH_2_-CH=)	---	---	---	---	---	---	1.1 ± 0.9	2.0 × 10^−2^
DHA (22:6, ω-3)	2.42 (m, -CH_2_-CH_2_-CO)	---	---	---	---	−1.0 ± 0.9	3.4 × 10^−2^	---	---
LA (18:2, ω-6)	2.77 (t, =CH-CH_2_-CH=)	---	---	---	---	---	---	1.0 ± 0.9	4.9 × 10^−2^
PUFAS	2.83 (m, =CH-CH_2_-CH=)	---	---	---	---	---	---	0.9 ± 0.9	4.9 × 10^−2^
UFAs	5.36 (m, -HC=CH-)	---	---	---	---	---	---	1.0 ± 0.9	4.9 × 10^−2^
PtdCho	3.32 (s, -N^+^(CH_3_)_3_)	---	---	---	---	---	---	1.2 ± 0.9	2.4 × 10^−2^
All GPLs	3.95 (m, PO-(3-CH_2_))	---	---	---	---	---	---	1.1 ± 0.9	4.9 × 10^−2^
TGs	4.29 (dd, 1-CH_2_/3-CH_2_ glyc.)	---	---	---	---	(0.7 ± 0.9)	(4.1 × 10^−2^)	---	---
Ceramide 1 δ 5.54 ^d^	5.54 (m, CH_2_)	---	---	1.0 ± 0.9	3.4 × 10^−2^	---	---	---	---
U1 ^e^	0.84 (br, weak)	---	---	---	---	---	---	−1.3 ± 0.9	1.4 × 10^−2^
U2 ^e^	1.03 (dd, weak)	---	---	---	---	---	---	−1.4 ± 0.9	7.5 × 10^−3^

## Data Availability

The data presented in this study were submitted to the Metabolomics Workbench database, and ID information will be provided in due time.

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
