# Peer review of "Impact of Sea Warming and 17-α-Ethinylestradiol Exposure on the Lipid Metabolism of Ruditapes philippinarum Clams"

_ijms, 2023, doi:10.3390/ijms24119485_

Round 1

Reviewer 1 Report

The paper is a continuation of earlier work published in Science of The Total Environment (doi.org/10.1016/j.scitotenv.2023.162898) and the experimental design is the same. Both manuscripts investigated the impact of elevated temperature  and 17-α-ethinylestradiol exposure on

metabolism of Ruditapes philippinarum. In the previous study, polar components were assessed, while this study investigates the lipidic profile. Thus, despite some similarities, the study has some scientific novelty.

"In general, this is a well-described manuscript. The introduction gives relevant background, and the results are well discussed. The presentation of the results is of high quality."

I have only two suggestions for Authors:

1)      Add some details to 4.2. section. Furthermore, „the methanol-chloroform-water” is not a method.

2)      Conclusion is rather a description of the results. Furthermore, this section could be more compact.

Author Response

Dear Reviewer,

We have uploaded a PDF file with our itemized response to all the comments made to our manuscript. Please find it enclosed.

Best regards,

Ana Gil

Reviewer 2 Report

The authors reported the results of an NMR metabolomics study of lipophilic extracts of R. philippinarum clams exposed to the hormone pollutant 17-α-ethynylestradiol (EE2) at 17°C and 21°C in relation to ocean temperature increases with global warming.

Results revealed novel markers of stress mitigation: high levels of PtdCho, PUFA (or PtdCho/glycerophosphocholine, PtdCho/acetylcholine ratio), linoleic acid, and low ratios of PUFA/glycine.

While it is not clear how these molecular cytological changes, now, will affect the response to global warming in R. philippinarum clams (conservation or transformation of the species under environmental effects), the results of this study are well-deserved.

Author Response

Dear Reviewer,

I have uploaded our response to the comments made to our manuscript, for your consideration.

Best regards,

Ana Gil
